

# Vision-based approach to knee osteoarthritis and Parkinson's disease detection utilizing human gait patterns

Zeeshan Ali[1,*], Jihoon Moon[2,*], Saira Gillani[3], Sitara Afzal[4], Muazzam Maqsood[5] and Seungmin Rho[6]

[1] Research and Development Setups, National University of Computer and Emerging Sciences, Islamabad, Pakistan
[2] Department of Data Science, Duksung Women's University, Seoul, Republic of South Korea
[3] Department of Information Technology and Computer Science, University of Central Punjab, Lahore, Pakistan
[4] Department of Software, Sejong University, Seoul, Republic of South Korea
[5] Department of Computer Science, COMSATS University Islamabad, Attock, Pakistan
[6] Department of Industrial Security, Chung-Ang University, Seoul, Republic of South Korea
* These authors contributed equally to this work.

Corresponding authors
Muazzam Maqsood,
muazzam.maqsood@cuiatk.edu.pk
Seungmin Rho, smrho@cau.ac.kr

## ABSTRACT

Recently, the number of cases of musculoskeletal and neurological disorders, such as knee osteoarthritis (KOA) and Parkinson's disease (PD), has significantly increased. Numerous clinical methods have been proposed in research to diagnose these disorders; however, a current trend in diagnosis is through human gait patterns. Several researchers proposed different methods in this area, including gait detection utilizing sensor-based data and vision-based systems that include both marker-based and marker-free techniques. The majority of current studies are concerned with the classification of Parkinson's disease. Furthermore, many vision-based algorithms rely on human gait silhouettes or gait representations and employ traditional similarity-based methodologies. However, in this study, a novel approach is proposed in which spatiotemporal features are extracted *via* deep learning methods with a transfer learning paradigm. Following that, advanced deep learning approaches, including sequential models like gated recurrent unit (GRU), are used for additional analysis. The experimentation is performed on the publicly available KOA–PD–normal dataset comprising gait videos with various abnormalities, and the proposed model has the highest accuracy of approximately 94.81%.

## INTRODUCTION

In the present environment, employing only clinical measures for disease diagnosis is not the only option; other diagnostic measures can offer a disease diagnosis with lower cost and high reliability. Several diseases cause damage to the quality of life, and various musculoskeletal disorders (MSDs) and neurological disorders have become more prevalent, such as knee osteoarthritis (KOA) and Parkinson's disease (PD) (*Nandy & Chakraborty, 2017*; *Verlekar, Soares & Correia, 2018*). These disorders and diseases

strongly influence human gait patterns, motivating the research community to investigate their analysis or diagnosis *via* human gait patterns. Explicitly, gait analysis is the study of the human walking style to recognize individuals and detect various pathologies. Every person has unique walking style and patterns, and the gait cycle attributes generate a straightforward and reliable method to differentiate and recognize individuals and detect normal and diseased gaits (*Nandy & Chakraborty, 2017*). In different scenarios, the human walking style varies from the usual style; however, when such problems as MSDs and central nervous system problems are involved, they can influence human gait (*Verlekar, Soares & Correia, 2018*).

From the perspective of MSDs, KOA is one of the most prevalent disorders related to the knees, which creates ongoing discomfort and increases the swing phase of the individual during walking (*Gornale, Patravali & Manza, 2016*). The likelihood of knee-joint degradation appears to be higher than that of other joints. Surveys report that KOA has a very strong influence on the lives of individuals worldwide (*Cross et al., 2014*). Further, KOA usually appears around the age of 40 and affects 654.1 million people (*Cui et al., 2020*). Because the likelihood of KOA increases with age, older individuals have greater discomfort in the knee joint. Similarly, PD is considered the most rapidly spreading illness from the perspective of neurological diseases (*Mai, Deng & Tan, 2024*). This disease affects about 6 to 7 million individuals. Some studies have found age to be one of the essential risk factors in PD (*Hou et al., 2019*). Both conditions affect the gait owing to joint and postural instability. Instead of only affecting the gait, these conditions also influence the quality of life, causing severe problems (*Swift, 2012*; *Prenger et al., 2020*). These conditions can cost about €7,000 to €17,000 for PD and $330 billion for KOA to be cured, causing financial problems. Assessment of these conditions utilizing the most reliable and least costly method can greatly assist healthcare professionals in addressing such challenges.

In existing studies, various methods have been designed to analyze gait patterns to diagnose these abnormalities. The feature set includes kinematics, and sensor-based methods, including stride and step length, step breadth, and so on, to acquire gait patterns. In contrast, temporal aspects address the sequence of gait cycle events, such as cadence (*Kour & Arora, 2019*). Some techniques also involve wearable sensors to acquire human gait. However, wearable sensors might cause discomfort or fatigue or lead to operational costs of sensors and radiation exposure.

Moreover, in some studies, PD can be detected by employing one-dimensional convolutional neural networks (CNNs) in which gait features can be processed with foot signals (*El Maachi, Bilodeau & Bouachir, 2020*). In contrast, the vision method can be more effective because it does not require human cooperation with gait acquisition. Gait can be acquired with low-resolution videos even if the person stands far from the camera. Hence, some studies have exploited the vision-based approach; however, they have worked with conventional and similarity-based statistical methods, which are parametric and limited in terms of real-time assessment (*Kour, Gupta & Arora, 2022*; *Ortells, Herrero-Ezquerro & Mollineda, 2018*).

Deep learning techniques are an excellent fit for the proposed research study as well because they have shown improved performance across a variety of application areas (*Bukhari et al., 2020*, *2022*; *Ansari et al., 2024*; *Maqsood et al., 2021*). Hence, we exploit an end-to-end deep learning model utilizing a transfer learning concept capable of exploiting spatiotemporal features. The proposed model works on segmented videos that can be acquired by employing the mask region-based CNN (Mask-R-CNN). The proposed model is validated on three class abnormalities instead of only focusing on PD as in previous methods (*Kour & Arora, 2019*); hence, it is a more complete solution. The proposed end-to-end solution can take video sequences of the gait cycle and extract spatiotemporal features utilizing state-of-the-art CNN variants, namely VGG16, MobileNet, and DenseNet, followed by computing the temporal dependencies of the gait cycle in the frames utilizing the gated recurrent unit (GRU) model. The experimentation is performed with the KOA–PD–normal (NM) dataset comprising three classes: KOA, PD, and healthy gait patterns. Moreover, the motivation behind this study is to present an innovative and cost-effective approach to PD and KOA disease detection by exploiting human gait with the use of spatiotemporal features. More precisely, in today's healthcare environment of musculoskeletal disorders (MSDs) and neurological disorders, diagnosing the diseases through only clinical tests is not the only option. The growing desire for low-cost and reliable testing methods has prompted the investigation of alternate methodologies. Secondly, musculoskeletal diseases (MSDs) and neurological diseases are becoming more common in the modern world, and they have a significant influence on people's gait or walking patterns. This has caused an increased interest among researchers in using human gait as a diagnostic tool. On the other hand, the rise of deep learning and computer vision algorithms provides the potential to use automated examination of gait for more precise and objective diagnosis of these diseases. The following are the pin-point contributions of this research:

1) an end-to-end deep learning model is proposed to detect KOA, PD, and healthy gait;
2) the proposed model acquires spatiotemporal gait features directly from silhouettes;
3) the suggested method is less costly because it utilizes low-resolution videos rather than costly sensors, and it has advantages over disease identification based on clinical measures, which is limited because of the scarcity of medical data.

The subsequent sections present the literature review, describe the proposed methodology, and present the results along with an analysis, before drawing the conclusions. Figure 1 illustrates list of classes available in the KOA–PD–NM dataset.

## LITERATURE REVIEW

In existing research, numerous approaches have been designed to exploit human gait patterns to detect PD and other abnormalities. Researchers have employed the disease detection procedure for many years *via* diverse methods or modalities; however, detecting such abnormalities utilizing gait is another viable and feasible option.

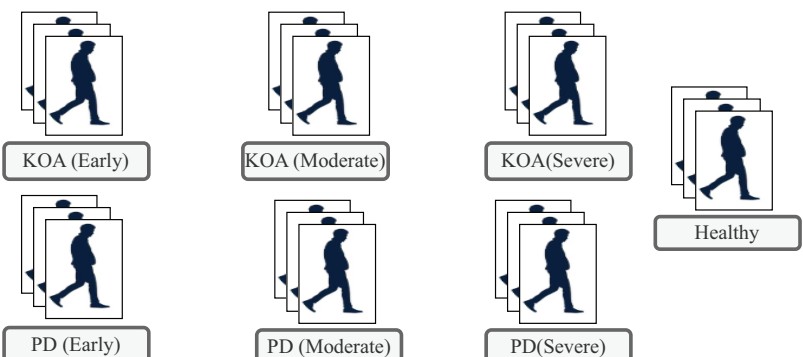

**Figure 1** **Types of classes available in the KOA-PD-NM dataset.**

## Sensor-based gait acquisition for PD detection

Research studies have acquired human gait patterns utilizing sensors in the context of PD detection utilizing human gait. The system's success depends on the quality of acquired data; hence, extreme caution must be maintained when collecting data.

Initially, the foundation of Geissler tubes in 1800 was a good method to examine the joint angles, but this led to considerable computational time (*Sutherland, 2002*). Some sensor-based methods (*e.g.*, the method by *Alam et al. (2017)*) applied the shoe sensor-based technique utilizing the support vector machine (SVM) algorithm to detect abnormal gait patterns and achieved an accuracy of 93.10%, a sensitivity of 93.10%, and a specificity of 94.10%. Similarly, *Sadeghzadehyazdi, Batabyal & Acton (2021)* proposed a deep learning model in which skeletal data captured with a Kinect sensor are employed to draw spatiotemporal features and classify the gait abnormalities. The proposed model is validated on the walking gait dataset and achieves up to 90.57% accuracy.

Further, *Vidya & Sasikumar (2022)* proposed a hybrid deep learning model involving CNNs and long short-term memory (LSTM) to predict PD and disease severity. They utilized the freely accessible PhysioNet gait dataset, comprising vertical ground reaction force signals from three walking experiments. The foot-sensitive resistors were affixed to each foot, and gait data were collected as the individuals walked at their normal speed. Likewise, *Shetty & Rao (2016)* proposed a PD detection framework in which gait time-series data are employed as a feature vector followed by a classifier called the Gaussian radial basis function kernel based on the SVM. The findings indicate that seven feature vectors are suitable for detecting PD with an accuracy of 75.00%. Following on, *Wahid et al. (2015)* suggested employing machine learning algorithms, including random forest and SVM. They utilized quantitative gait features acquired with sensors as input to diagnose PD. Over various experiments, the spatiotemporal features commonly involved stride length, step length, step width, cadence, double support time, stance time, swing time, step time, and stride time. Their work achieved an accuracy of 92.60% utilizing random forest and 80.40% with the SVM. *Li & Li (2022)* also proposed machine learning classifiers that involve logistic regression, SVM, decision trees, and k-nearest neighbors

(KNN) to diagnose patients and healthy controls. The input of these algorithms is the gait features recorded with force sensors. The experiment was performed with the Gait PD dataset available on PhysioNet. *Wang, Zeng & Dai (2024)* proposed frameworks based on machine learning algorithms to classify gait to detect PD and its severity. Their method is based on signal processing, utilizing time-series data on the vertical ground reaction force. To demonstrate the value of their method, they employed a gait database containing data on 93 patients with PD and 73 healthy subjects. The highest classification method exhibits performance of approximately 98.20% and 96.69% utilizing SVMs.

Some advanced models such as *Nguyen et al. (2022)* proposed transformer networks by exploiting the one-dimensional data of gait features to detect PD. These features were acquired by a sensor attached to the person's foot. They employed dimension-reduction techniques to overcome the memory problems of the transformer models. Their suggested method outperforms the existing methods with an accuracy of 95.20% on the PhysioNet database.

As in most of these studies, sensor-based gait acquisition is employed; however, one possible limitation of this acquisition is the cost of sensors. Moreover, this method necessitates that a person cooperates with the system, such as wearing foot pressure sensors.

## Sensor-based KOA detection utilizing gait

Other than PD, some wearable sensor methods help determine other conditions, such as KOA, utilizing gait (*He, Liu & Yi, 2019*). In this context, *Kwon et al. (2020)* proposed a KOA detection system in which features are extracted from radiographic images of X-rays by employing a deep learning model, namely inception-ResNet-V2. Following on, gait features such as the ankle joint, stride, and others are combined for classification using SVM, followed by KOA disease grading. Their proposed model produces good results with an F1 score of 71.00%, sensitivity of 70.00%, and precision of 76.00%. The findings of this research indicate that in addition to radiographic images, gait data play a complementary role with respect to KOA classification.

Likewise, *Kotti et al. (2017)* suggested a technique to assess the validity of a rule-driven strategy for comparing 47 participants with KOA *versus* healthy participants utilizing gait data collected from floor sensors. Similarly, *Tereso, Martins & Santos (2015)* exploit the use of inertial sensors to acquire the spatiotemporal parameters of gait to identify KOA abnormalities. This sensor-based modality works well yet is limited by variables that include high cost, time and energy consumption, and wearing difficulty.

## Vision-based methods for KOA and PD utilizing gait

Some vision-based methods diagnose these diseases utilizing a human gait in which gait data are recorded with a camera. In this context, *Kour, Gupta & Arora (2022)* presented a vision-based approach to classify human walking patterns as normal, PD, and KOA. Various gait features (*e.g.*, kinematics, motion, and other statistical methods) are extracted, and classification is performed with the KNN algorithm. Fractional-order Darwinian particle swarm optimization (PSO) is also used to increase the detection method's accuracy

and reliability for retrieving regions of interest. Their suggested technique yields 90.00% accuracy, 85.00% sensitivity, 90.44% specificity, and 89.66% precision.

Likewise, some camera-based methods, including digital and charge-coupled device cameras, are also utilized in studies, employing the conventional methods of principal component analysis (PCA), linear discriminant analysis (LDA), sequential backward selection (SBS) Hough transform, kernel principal component analysis (KPCA), and others (*Chen et al., 2011*). These methods employ the gait features of speed, energy, swing time, and joint angles. Similarly, *Verlekar, Soares & Correia (2018)* utilized a two-dimensional video-based system in which feet-related features, such as the step length and fraction of the foot, and body-related features with a human gait energy image, including the center of gravity, are employed to detect gait abnormalities. They applied a classification approach based on classic machine learning methods, such as the SVM. The findings indicate that the precision of the center of gravity features is affected by low-resolution videos.

From the above literature, the detection of PD and KOA utilizing gait patterns has primarily been exploited with sensor-based, kinematic, and handcrafted features; however, the classification of these conditions utilizing vision with deep learning has not been adequately explored. For instance, in one study (*Kour, Gupta & Arora, 2022*), although the vision-based approach is exploited, the authors employed conventional methods, specifically a marker-based vision approach, necessitating a proper laboratory setup and requiring more space and time to acquire the gait. A crucial research question arises here: what if we diagnose these diseases (*i.e.*, *via* a joint framework for diagnosing both diseases) with deep learning methods along with a transfer learning paradigm? Moreover, what if we employed only vision data that are not high quality or resolution, disregarding other features and laboratory environmental setups?

## MATERIALS AND METHODS

This section explains how the proposed model works step by step. Initially, the video segmentation to acquire silhouettes is conducted with the Mask-R-CNN model followed by the VGG-GRU, MobileNet-GRU, and DenseNet-GRU models, designed with the transfer learning paradigm to extract spatiotemporal features from the walking style to classify the gait as PD, KOA or normal. Figure 2 illustrates the proposed work.

### Datasets

This research employs the KOA–PD–NM gait dataset to detect PD and KOA based on human gait (*Kour, Gupta & Arora, 2020*). This dataset accounts for various factors in collected gait videos, such as age, gender, and severity levels. For each disease (KOA or PD), videos from varying severity levels are provided, such as early, moderate, and severe. The dataset consists of 96 subjects, including 50 individuals with KOA, 16 with PD, and 30 healthy individuals. There are two sequences for every person, one from left to right and the other from right to left, and all videos are in the format of .mov. Individuals' faces are not revealed in the videos and are blurred during recording. Moreover, this dataset can also be used in existing studies (*Kour, Gupta & Arora, 2022*, *2024*; *Khessiba et al., 2023*;

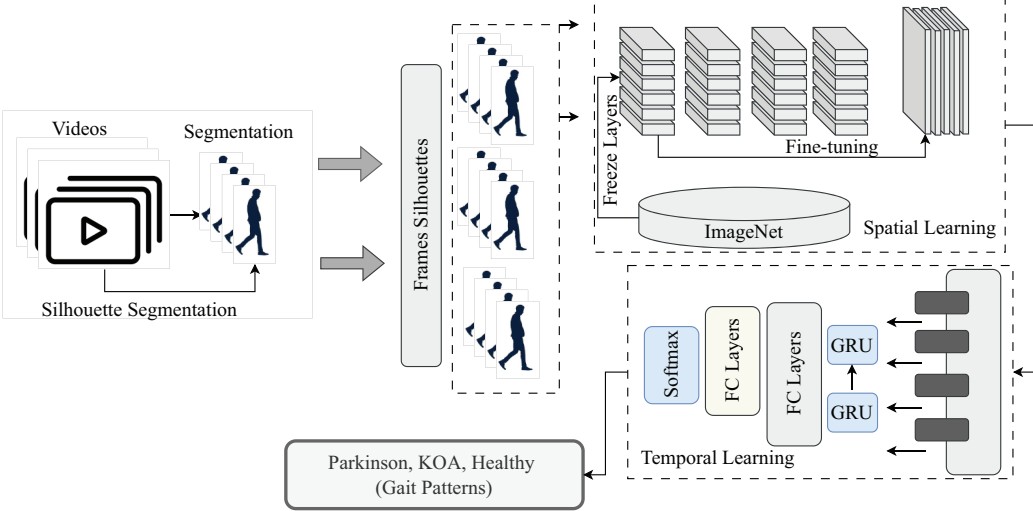

**Figure 2  A pictorial overview of proposed methodology for detection of normal and abnormal gait patterns.**               

*Hassine et al., 2024*; *Martinel, Serrao & Micheloni, 2024*; *Kour, Sunanda & Arora, 2022*; *Sousse, Grassa & Manzanera, 2023*). More detailed statistics of the dataset are listed in Table 1.

## Silhouette segmentation

This research exploits the gait features directly from walking captured utilizing a video camera. Hence, after acquiring videos of healthy and PD or KOA patients from dataset, we segment these silhouettes to extract the region of interest indicating the human gait cycles over several video frames. Silhouette segmentation is one of the most widely utilized gait representations to indicate gait patterns. Hence, to acquire the silhouette gait representation of a person, we fine-tuned the Mask-R-CNN (*Kaiming et al., 2017*) model on the Pedestrian Detection Database (PennFudan) based on the Faster-R-CNN architecture with an additional branch of segmentation. This architecture is based on two-stage object detection comprising a region proposal network for the proposal of regions and person detection and segmentation.

## Spatiotemporal gait patterns from segmented videos

After silhouette segmentation for every walking sequence in the dataset, the stack of silhouettes is utilized to extract spatiotemporal gait features utilizing the VGG16, MobileNet, DenseNet, and GRU models. The fundamental rationale for employing these models is that, while the CNN and its variants are extensively employed for gait classification owing to their ability to extract informative and spatial information, they cannot investigate temporal relationships between gait cycles. In this case, the GRU model is employed to capture the temporal dependencies of the gait cycle (*i.e.*, frame by frame in sequences). Spatiotemporal models are utilized with a transfer learning paradigm to classify gait patterns as PD, KOA, or normal. The models are described in detail below:

**Table 1 Details of the dataset.**

| No. | Classes | Severity | Total videos |
|---|---|---|---|
| 01 | KOA | Early | 30 |
| 02 | KOA | Moderate | 40 |
| 03 | KOA | Severe | 30 |
| 04 | PD | Early | 12 |
| 05 | PD | Moderate | 14 |
| 06 | PD | Severe | 5 |
| 07 | Healthy | – | 60 |
| 08 | Total | – | 191 |

### VGG16+GRU

The segmented video frames with dimensions of $224 \times 224 \times 3$ are passed as input to the VGG16 (*Simonyan, 2014*) to extract spatial or semantic information from the segmented frames to acquire the spatiotemporal gait features. More precisely, VGG16 is a state-of-the-art CNN based on deepening the architecture of the network with small convolutional filters (*i.e.*, $3 \times 3$). The VGG16 model is a much more effective convolutional network architecture that reaches state-of-the-art accuracy on ImageNet Large Scale Visual Recognition Challenge (ILSVRC) classification and localization tasks while being adaptable to other image recognition datasets. This model introduces more convolutional layers in the network by adding small filters. For instance, a stack of three convolutional layers capturing a small receptive field is equivalent to one $7 \times 7$ filter-based convolutional layer. This concept also introduces more nonlinearity and computational overload of parameters. We employed this architecture trained on ImageNet weights to extract features from the video frames and fine-tune the model with the transfer learning concept.

Temporal features are exploited with the GRU model, where a sequence of spatial features holding the semantic image information is passed to exploit human walking patterns over time. The GRU model is an extended version of recurrent neural networks (RNNs) and long short-term memory (LSTM) networks capable of dealing with the long-term dependencies of gait cycles, frame by frame. The distinction between GRU and LSTM is that GRU eliminates the cell state, employing the hidden state to convey information. This method comprises two gates: the reset and update gates. The working of the update gate is identical to the forget and input gates in the LSTM network, determining what information to discard and what novel information to include.

Similarly, the reset gate determines which information from the past is discarded and forgotten. Because GRUs have fewer tensor operations, they can be trained faster than LSTM networks. The proposed model employs two GRU layers comprising 16 and eight hidden units. The final feature set is passed to fully connected layers to classify the underlying walk as a normal or abnormal gait (KOA or PD). Figure 3 shows a detailed view of the proposed model for classifying normal and abnormal gait patterns.

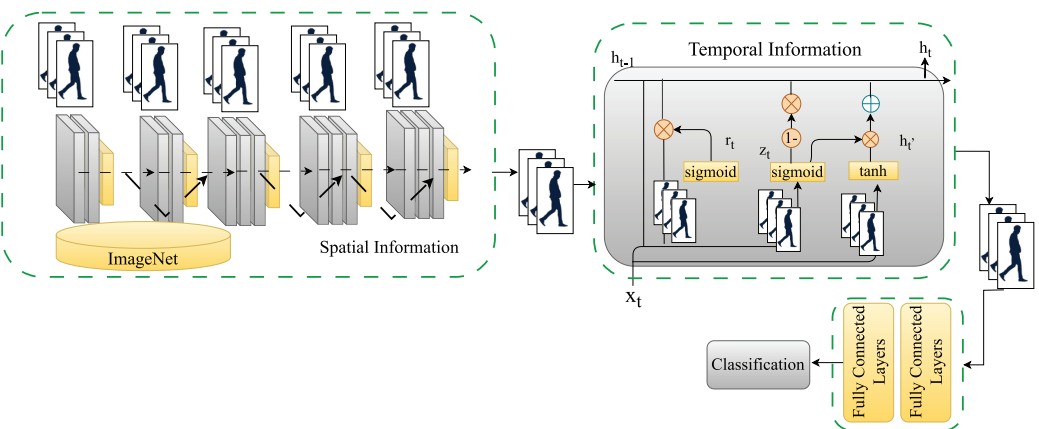

**Figure 3 A detailed view of the proposed model for classifying normal and abnormal gait patterns.**

### MobileNet+GRU

Similar to the previous model of VGG16+GRU, we proposed another, namely MobileNet+GRU, where we acquire the spatiotemporal gait features from the segmented video frames with the dimensions of 224 × 224 × 3. These frames are passed as input to the MobileNet (*Howard, 2017*) to extract the spatial or semantic information from the segmented frames. The MobileNet family of CNNs is an efficient class of deep learning models, particularly for mobile and embedded vision applications. MobileNet architectures are based on the simplest architecture comprising depth wise separable convolutions to make them lightweight. These convolutions are conventional convolutions factorized into a 1 × 1 depth wise convolution, also referred to as a pointwise convolution. Each input channel is subjected to one filter in the depth wise convolution. The depth wise convolutional outcomes are combined utilizing a 1 × 1 pointwise convolution. This factorization results in significant reductions in processing and model size. We employed this architecture trained on ImageNet weights to derive features from video frames and fine-tune the model with the transfer learning technique.

Similar to the previous model of VGG16+GRU, we also exploited temporal information utilizing one of the improved variants of RNNs, namely GRUs. The suggested model employs two GRU layers with 16 and eight hidden units, respectively. Beyond that, the final feature set is given to fully connected layers to classify the underlying walk as normal or abnormal (KOA or PD).

### DenseNet+GRU

Similar to the previous two models, a DenseNet-based model is also proposed *i.e.*, DenseNet+GRU. The DenseNet model (*Huang et al., 2017*) is also a state-of-the-art CNN. In this architecture, each layer is connected to the next in a feed-forward manner. Further, DenseNet offers numerous appealing benefits: it solves the vanishing-gradient problem, improves feature propagation, increases feature reusability, and significantly minimizes the

number of parameters. Mathematically, the *lth* layer receives feature maps of the previous layers as input:

$$X_l = H_l([x_0, x_1, \ldots x_{l-1}]). \tag{1}$$

The term $[x_0, x_1, \ldots x_{l-1}]$ refers to the concatenation of the feature maps of previous layers. The model is called DenseNet because of its dense connectivity between the layers. In this research, we utilized this architecture trained on ImageNet weights to generate features from video frames and fine-tune the model via the transfer learning approach. Moreover, DenseNet easily blends the benefits of identity mappings, deep oversight, and diverse depths while adhering to the basic connection rule.

Similar to prior models (*i.e.*, VGG16+GRU and MobileNet+GRU), we also exploited the temporal information with one of the improved variants of RNNs (*i.e.*, GRUs). The proposed model consists of two GRU layers, each with 16 and eight hidden units. The final feature set is then processed through fully connected layers to determine if the gait is normal or abnormal (KOA or PD).

## Hyperparameters and implementation details

Hyperparameters play a major role in fine-tuning the performance of deep models. The hyperparameters of the models include the learning rate (set to 0.01) and the weight optimizer (set to Adam). The models run for 100 epochs; however, early stopping callbacks are set based on the validation loss to prevent overfitting. Likewise, a dropout rate of 0.3 is set. Frame sizes are set to $224 \times 224 \times 3$, and the sequence length of frames is set to 100. Moreover, the implementation details include the hardware/software platforms; hence, all implementation is conducted in the Python language. Simulations are run on Google Colab with an Nvidia Tesla K80 graphics processing unit (GPU) with 12 GB of video random access memory (VRAM). These Colab notebooks are operated by a virtual machine running Ubuntu Linux as an operating system.

## EXPERIMENTS AND DISCUSSION

This section explains the dataset and evaluation metrics to validate the performance of the proposed model. The results of the proposed method are discussed.

## Evaluation metrics

The metrics employed to evaluate the model's performance include accuracy, precision, recall, and the F1 score which are considered to be the standard metrics for classification problem in machine learning. The rationale of utilizing these metrics is to indicate the comprehensive performance of proposed model in detecting normal and abnormal gait patterns associated with KOA and PD. The following mathematical equations evaluate the performance of the model:

$$Accuracy = \frac{TP + TN}{TP + TN + FP + FN} \tag{2}$$

$$Precision = \frac{TP}{TP + FP} \tag{3}$$

$$\text{Recall} = \frac{TP}{TP + FN} \tag{4}$$

$$\text{F1 score} = 2.\frac{\text{Precision.Recall}}{\text{Precision} + \text{Recall}}, \tag{5}$$

In the above Eqs. (2)–(5), TP indicates the true positives, TN denotes true negatives, FP represents false positives, and FN indicates the false negatives.

## Experiments and results

This section examines the suggested model's performance on a comprehensive dataset. The performance is evaluated using classification accuracy, which demonstrates its ability to discriminate between normal and pathological gait patterns. As mentioned in section "Materials and Methods", silhouettes are extracted from the raw video frames. Some samples of video frames and their corresponding segmented silhouettes are depicted in Fig. 4. Furthermore, Fig. 4 depicts silhouettes of different gait patterns, including KOA (first row), PD (second row), and normal (third row). As observed from the silhouettes, the head is blurry because it is intentionally hidden in the original frames during recording to anonymize the identities of participants for data collection. After the extraction of silhouettes, frame by frame, if these silhouettes are again combined to form the videos, it would result in segmented videos. To address the limited data problem, we augmented the training frames of the sequences with additive Gaussian noise (*i.e.*, augmentation that does not disrupt the gait cycle patterns).

Following on, the segmented videos are input into the proposed models, including VGG16+GRU, MobileNet+GRU, and DenseNet+GRU, to acquire the spatiotemporal gait patterns of subjects walking in a video to classify their gait as abnormal (KOA or PD) and normal. Initially, we tested the algorithm with three classifications: KOA, PD, or healthy. The training and testing videos were divided with a ratio of 80:20. Moreover, we shuffled the dataset five times for each model to accurately assess the performance of the model. Table 2 presents the results of the VGG16+GRU model, revealing that the model performs very well in different runs with the random shuffling of the training and testing datasets. The highest metrics that the model achieves are 83.78% accuracy, 59.00% precision, 64.00% recall, and 61.00% F1 score. Table 2 also provides the mean ± standard deviation for all simulations. The overall values of the mean accuracy, precision, recall, and F1 score of the proposed VGG16+GRU model to detect gait abnormalities are 80.54%, 56.40%, 61.00%, and 58.40%, respectively.

After validating the results of the proposed VGG16+GRU model, the same experimental setup was employed for MobileNet+GRU. The training and testing videos were divided like in the previous scenario, with an 80:20 ratio. Furthermore, we shuffled the dataset five times for each model to correctly assess its performance. Table 3 presents the findings of the MobileNet+GRU model, demonstrating that the model works well in different runs with random shuffling of the training and testing datasets. The model's highest metrics are 83.78% accuracy, 59.00% precision, 64.00% recall, and 61.00% F1 score. Table 3 also includes the mean × standard deviation. The low deviation in the accuracy,

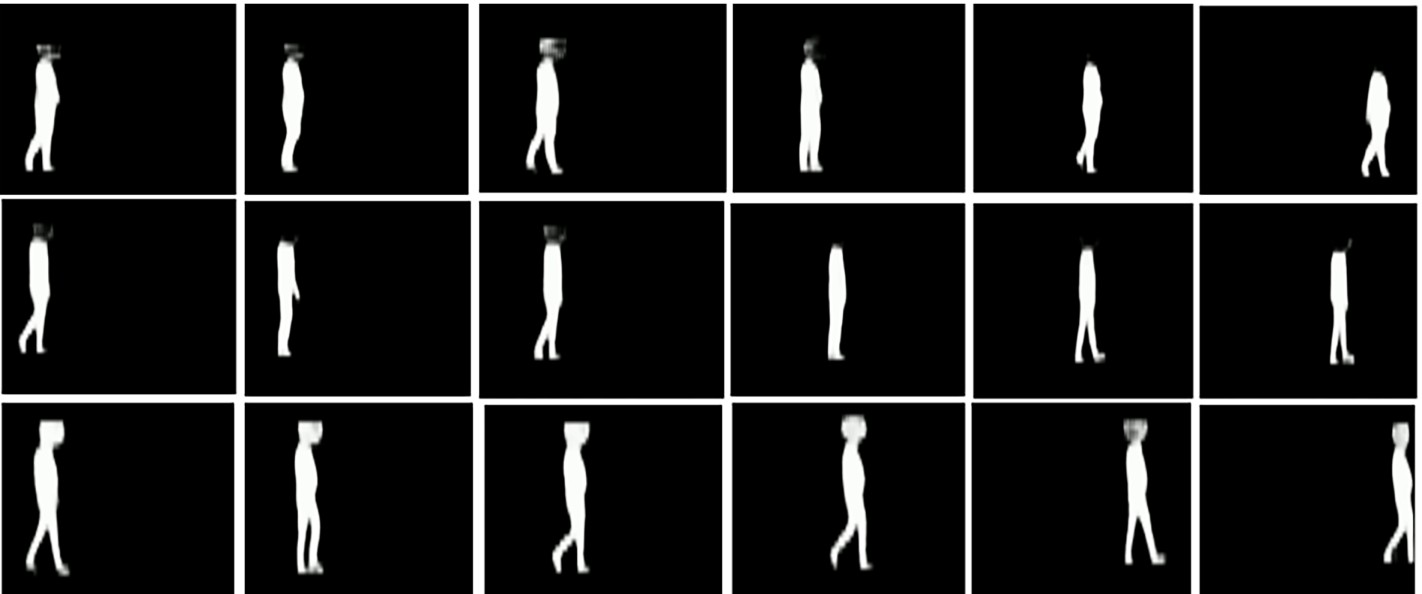

**Figure 4 Samples of silhouettes segmentation of gait videos.** (A) First row corresponds to KOA disease subjects (B) Second row corresponds to PD diseased subjects (C) Third row corresponds to healthy subjects.               

**Table 2 Results of the proposed VGG16+GRU with multi-class classification.**

| No. | Accuracy (%) | Precision (%) | Recall (%) | FScore (%) |
|---|---|---|---|---|
| 01 | 78.37 | 57.00 | 63.00 | 59.00 |
| 02 | 83.78 | 82.00 | 73.00 | 75.00 |
| 03 | 73.00 | 51.00 | 63.00 | 57.00 |
| 04 | 72.90 | 72.00 | 86.00 | 71.00 |
| 05 | 83.70 | 77.00 | 80.00 | 78.00 |
| Mean ± Std. dev | 78.35 ± 4.83 | 67.80 ± 11.86 | 73.00 ± 9.14 | 68.00 ± 8.49 |

precision, recall, and F1 score performance metrics indicates that MobileNet+GRU exhibits good results.

The mean overall accuracy, precision, recall, and F1 score values attained by this model are about 81.08%, 57.00%, 61.60%, and 58.60%, respectively. However, when comparing MobileNet+GRU and VGG16+GRU, the results of MobileNet+GRU are slightly better than VGG16+GRU in terms of accuracy, precision, recall, and the F1 score.

As with the prior two models, DenseNet+GRU is also validated with the same experimental settings. Table 4 presents the results of the DenseNet+GRU model. The results indicate that the model performs admirably across runs with random shuffling of the training and testing datasets. The model's highest metrics are 83.78% for accuracy, 59.00% for precision, 64.00% for recall, and 61.00% for the F1 score. The overall

**Table 3  Results of the proposed MobileNet+GRU with multi-class classification.**

| No. | Accuracy (%) | Precision (%) | Recall (%) | FScore (%) |
|---|---|---|---|---|
| 01 | 78.00 | 57.00 | 63.00 | 59.00 |
| 02 | 81.00 | 79.00 | 71.00 | 73.00 |
| 03 | 75.68 | 51.00 | 65.00 | 57.00 |
| 04 | 78.38 | 53.00 | 57.00 | 55.00 |
| 05 | 86.48 | 59.00 | 67.00 | 62.00 |
| Mean ± Std. dev | 79.91 ± 3.69 | 59.80 ± 10.01 | 64.60 ± 4.63 | 61.20 ± 6.34 |

**Table 4  Results of the proposed DenseNet+GRU with multi-class classification.**

| No. | Accuracy (%) | Precision (%) | Recall (%) | FScore (%) |
|---|---|---|---|---|
| 01 | 81.08 | 91.00 | 68.00 | 68.00 |
| 02 | 83.78 | 90.00 | 73.00 | 76.00 |
| 03 | 78.38 | 53.00 | 67.00 | 59.00 |
| 04 | 83.78 | 60.00 | 65.00 | 62.00 |
| 05 | 83.00 | 56.00 | 64.00 | 59.00 |
| Mean ± Std. dev | 82.00 ± 2.06 | 70.00 ± 16.89 | 67.40 ± 3.14 | 64.80 ± 6.49 |

performance for accuracy, precision, recall, and the F1 score with mean plus or minus the standard deviation is about 80.54%, 57.00%, 61.00%, and 58.40%, respectively. If the performance of all models is contrasted, the performance of MobileNet+GRU is slightly better than that of the other two models. In addition, the training and testing plots for all three models are also presented below, indicating the accuracy and loss values over a set of epochs. The epochs were set initially to 100; however, a callback is also set to stop early if the validation loss does not improve. Figure 5 indicates that the proposed model exhibits good convergence over epochs and reaches the optimal values of accuracy and loss.

Subsequently, another experimental scenario is also implemented in which the KOA and PD classes are combined into a single abnormal class, with the other class representing healthy gait patterns, resulting in a binary classification challenge. The reason for accessing the models in this way is the limited number of videos for PD abnormalities in the dataset, as indicated in Table 1. Second, it offers another experimental test to determine the performance from a different perspective. In this case, we kept the experimental settings from the previous scenarios (*i.e.*, the training, testing, and shuffling of datasets).

Tables 5–7 present the results of all three models. For each model, we shuffled the datasets and tested the model's generalizability in detecting abnormal and normal gait patterns. The mean plus or minus the standard deviation of all three models was also computed. Table 5 reveals that VGG16+GRU has the highest accuracy, precision, recall, and F1 score at about 99.00% for all four metrics. Similarly, the mean plus or minus the standard deviation of the accuracy, precision, recall, and F1 score of the VGG16+GRU

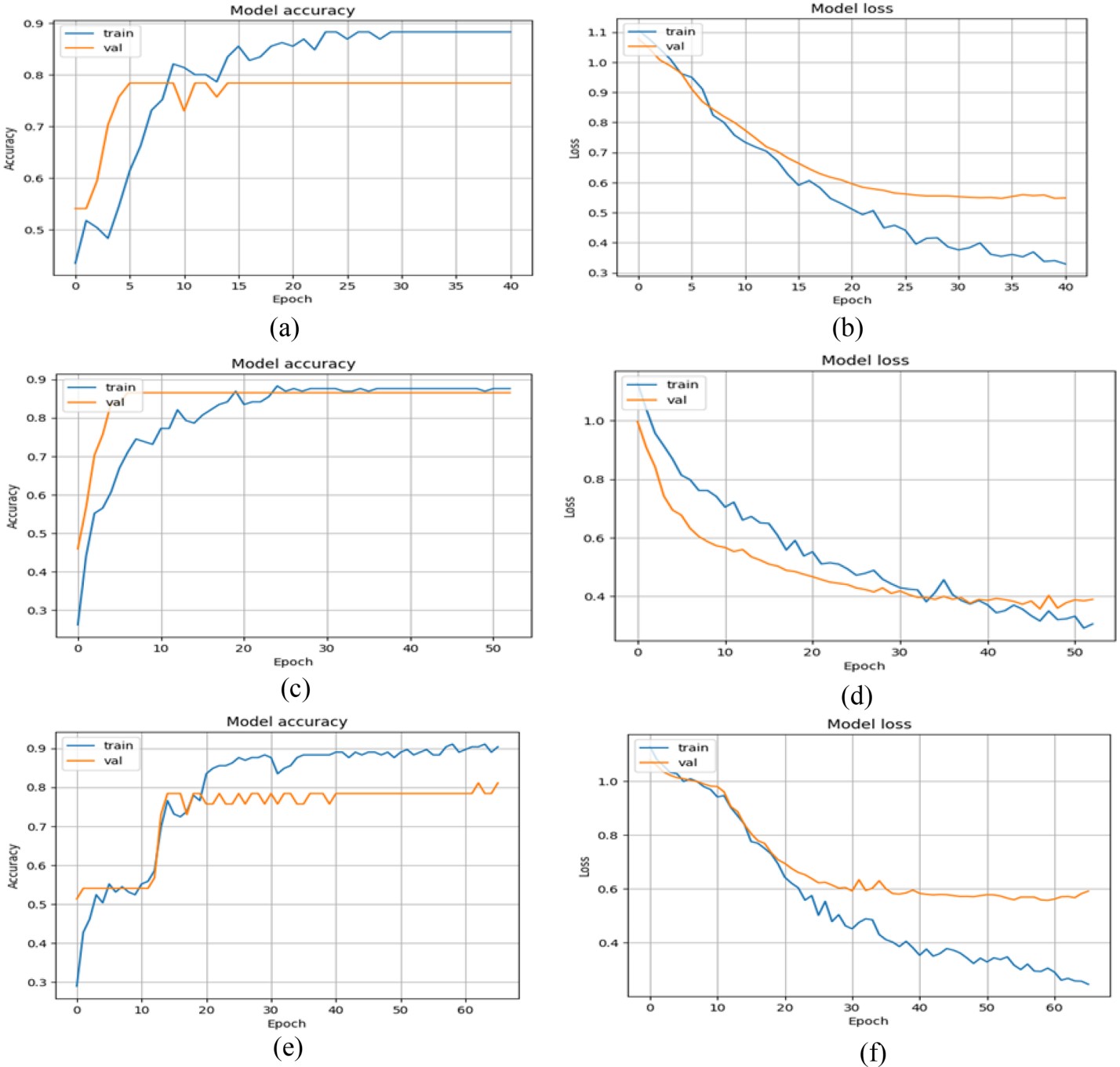

**Figure 5 Training and validation graphs of accuracy and loss values with multi-class classification.** (A and B) Accuracy and loss curves of VGG16+GRU (C and D) Accuracy and loss curves of MobileNet+GRU (E and F) Accuracy and loss curves of DenseNet+GRU.

**Table 5  Results of the proposed VGG16+GRU with binary classification.**

| No. | Accuracy (%) | Precision (%) | Recall (%) | FScore (%) |
|---|---|---|---|---|
| 01 | 99.00 | 99.00 | 99.00 | 99.00 |
| 02 | 94.59 | 94.00 | 94.00 | 94.00 |
| 03 | 91.89 | 91.00 | 92.00 | 92.00 |
| 04 | 94.59 | 93.00 | 93.00 | 93.00 |
| 05 | 94.00 | 97.00 | 90.00 | 93.00 |
| Mean ± Std. dev | 94.81 ± 2.32 | 94.80 ± 2.86 | 93.60 ± 3.01 | 94.20 ± 2.48 |

**Table 6  Results of the proposed MobileNet+GRU with binary classification.**

| No. | Accuracy (%) | Precision (%) | Recall (%) | FScore (%) |
|---|---|---|---|---|
| 01 | 97.30 | 96.00 | 98.00 | 97.00 |
| 02 | 94.59 | 94.00 | 94.00 | 94.00 |
| 03 | 89.19 | 88.00 | 88.00 | 88.00 |
| 04 | 91.89 | 90.00 | 92.00 | 91.00 |
| 05 | 98.10 | 93.00 | 96.00 | 94.00 |
| Mean ± Std. dev | 94.21 ± 3.33 | 92.20 ± 2.86 | 93.60 ± 3.44 | 92.80 ± 3.06 |

**Table 7  Results of the proposed DenseNet+GRU with binary classification.**

| No. | Accuracy (%) | Precision (%) | Recall (%) | FScore (%) |
|---|---|---|---|---|
| 01 | 99.02 | 99.01 | 99.00 | 99.00 |
| 02 | 91.89 | 92.00 | 91.00 | 91.00 |
| 03 | 86.49 | 85.00 | 86.00 | 85.00 |
| 04 | 97.00 | 98.00 | 94.00 | 96.00 |
| 05 | 98.00 | 98.01 | 98.00 | 98.00 |
| Mean ± Std. dev | 94.48 ± 4.69 | 94.40 ± 5.32 | 93.60 ± 4.76 | 93.80 ± 5.19 |

model is about 94.81 ± 2.32%, 94.80 ± 2.86%, 93.60 ± 3.01%, and 94.20 ± 2.48%, respectively. Likewise, the proposed MobileNet+GRU model results are also encouraging in this binary case, achieving the highest accuracy, precision, recall, and F1 score values of 98.10%, 93.00%, 96.00%, and 94.00%. The mean plus or minus the standard deviation of this model in the binary case in terms of accuracy, precision, recall, and F1 score is about 94.21 ± 3.33%, 92.20 ± 2.86%, 93.60 ± 3.44%, and 92.80 ± 3.06%. Similarly, in Table 7, DenseNet performs well, achieving an overall accuracy, precision, recall, and F1 score of about 99.00%. In this scenario, if the performance of all models is contrasted, then the overall VGG16+GRU outperforms the other two models: MobileNet+GRU and DenseNet

+GRU. In addition, the training history plots of the accuracy and loss over several epochs are also plotted for all three models in Fig. 6.

Furthermore, when model performance is analyzed depending on two experimental settings, the second experimental setting (*i.e.*, binary classification) has better results. The rationale for the weak performance in the first experimental scenario is that the model has trouble detecting abnormal gaits associated with KOA or PD. However, when combined, both are recognized as abnormal gait patterns, and the model precisely distinguishes them from normal gait. Further, the number of samples per severity level is relatively low, particularly in the case of PD. To address this, we augmented the frames of the sequences with additive Gaussian noise (*i.e.*, augmentation that does not disrupt the gait cycle patterns), yet the model still lags. This problem can be addressed by increasing the diversity of samples in the dataset. In addition, the receiver operating characteristic curve is also depicted in Fig. 7 with a high area under the curve score of 0.89, reflecting the better ability of the model to balance TP and FP rates, indicating its effectiveness in the underlying tasks.

## Discussion, analysis, and comparisons

Gait is a direct indication of one's health, driving the present-day attempts to streamline the study of abnormal gait to help clinicians make decisions (*Ortells, Herrero-Ezquerro & Mollineda, 2018*). Existing studies have employed several methods to exploit gait patterns in diagnosing diseases or abnormalities. However, most existing work focuses on sensor-based gait acquisition, including the stride and step length, step breadth, and so on, whereas temporal aspects deal with the sequence of gait cycle events, such as cadence (*Kour & Arora, 2019*).

Despite adopting a vision-based approach, some studies have employed statistical and conventional methods to solve this problem (*Kour, Gupta & Arora, 2022*; *Ortells, Herrero-Ezquerro & Mollineda, 2018*). Nevertheless, these models have not been investigated with state-of-the-art deep learning models. A question arises: how can existing conventional methods be automated *via* advanced methods? Hence, one of the significant contributions of this research is to exploit gait patterns in diagnosing abnormalities utilizing deep learning models capable of extracting spatiotemporal gait features directly from human silhouettes. The proposed technique is less expensive because it utilizes low-resolution videos rather than expensive sensors, offering advantages over clinical measure-based illness detection, which is limited owing to a lack of medical data. Moreover, this method does not require a proper laboratory setup, necessitating more space and time to acquire the gait, as in existing methods (*Kour, Gupta & Arora, 2022*).

These contributions are justified by the good results of the proposed deep learning models in diagnosing normal and abnormal gait patterns. The proposed model extracts the spatiotemporal features. We froze the initial layers of VGG16, MobileNet, and Dense-Net trained on ImageNet to draw features and fine-tune those features *via* a transfer learning strategy. These state-of-the-art CNNs extract spatial information indicating semantic information from silhouette images, such as a person's body posture and shape. During walking, the gait cycle temporal information is exploited with one of the variants of RNNs:

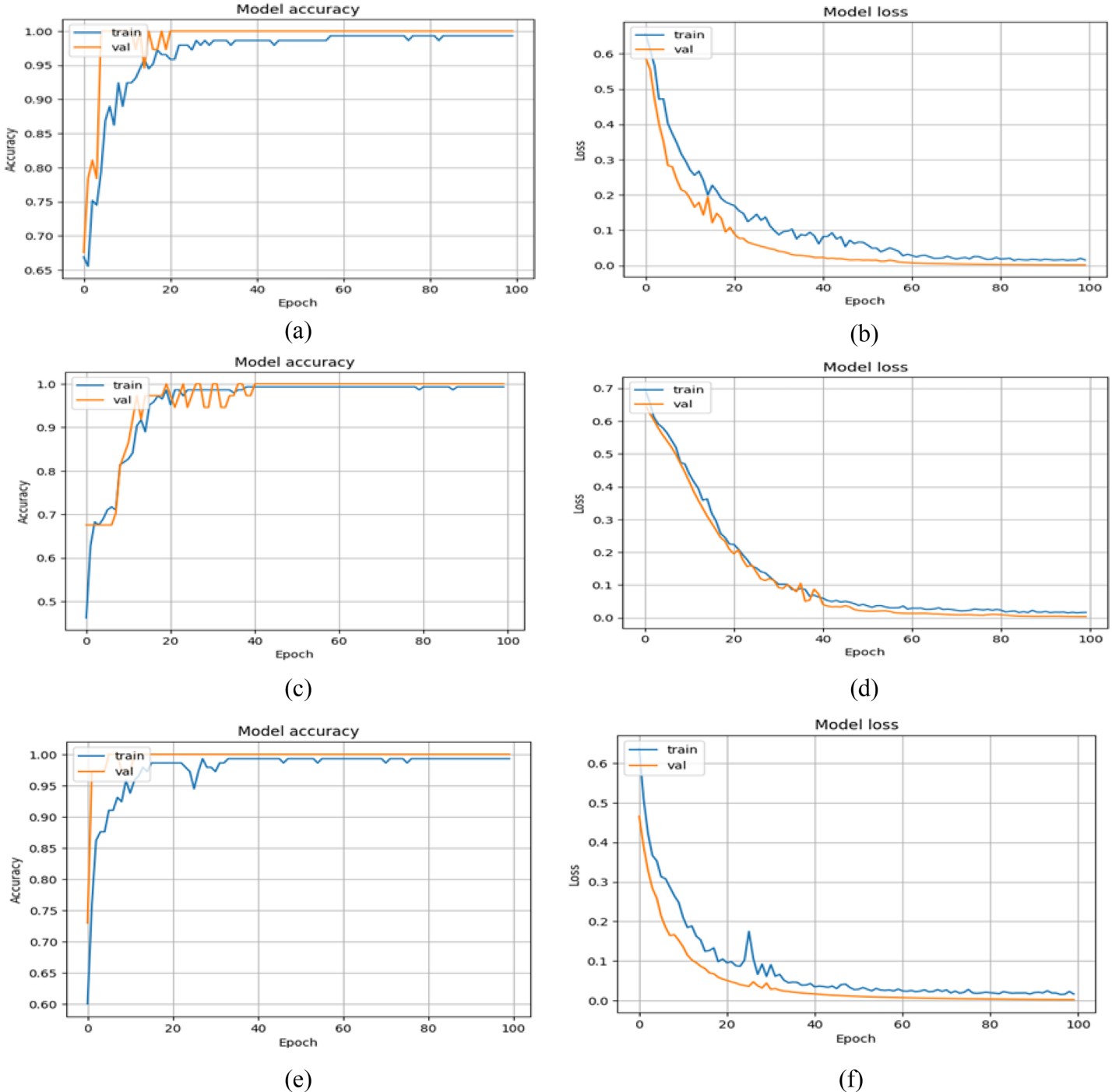

**Figure 6 Training and validation graphs of accuracy and loss values with binary classification setup.** (A and B) Accuracy and loss curves of VGG16+GRU (C and D) Accuracy and loss curves of MobileNet+GRU (E and F) Accuracy and loss curves of DenseNet+GRU.

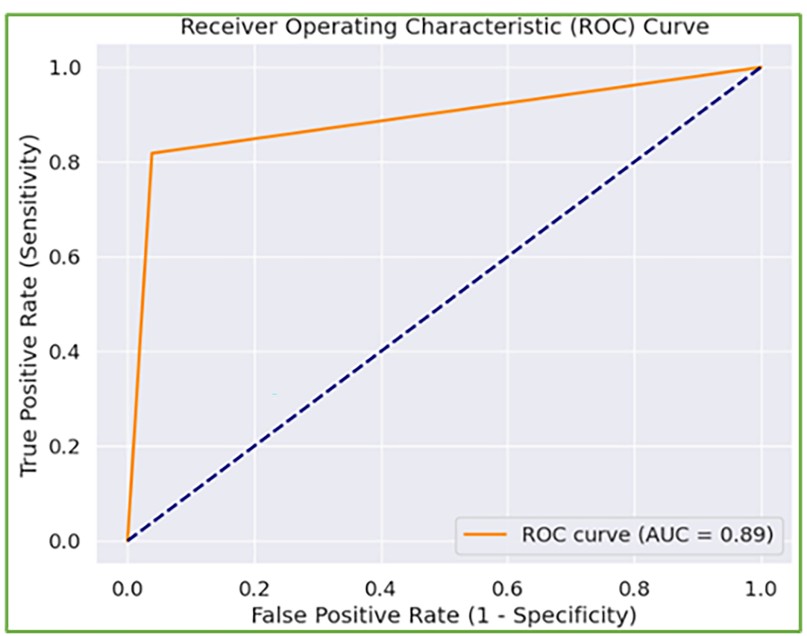

**Figure 7 Receiver operating characteristic (ROC) curve of the model identifying normal and abnormal gait.**

GRU. We tested the model in various scenarios: two- and three-class categorizations of gait patterns. In both scenarios, the proposed model displays good results. Although the suggested model is based on deep learning paradigms that are fundamentally black-box, it may be made more interpretable by including attention processes, feature attribution methods (*e.g.*, SHAP, Grad-CAM), and other explainability approaches.

To further demonstrate the worth of the proposed work, a comparison against existing methods is also conducted as shown in Table 8. In Table 8, the vision-based data are exploited with the KOA–PD–NM dataset in only two studies to classify the gait patterns as normal and abnormal. *Kour, Gupta & Arora (2022)* employed fractional-order Darwinian particle swarm optimization to segment regions of interest and extract gait features. Afterward, the KNN classifier was employed for the extracted feature values. Likewise, *Khessiba et al. (2023)* employed a graph convolutional network (GCN) in which nodes of the graph denote the body joint positions and achieve a good accuracy of 93.75%. Furthermore, unlike computationally demanding and sophisticated techniques such as PSO-based optimization and GCN-based advanced models, our method achieves great accuracy while using substantially less computational resources.

In addition to the vision-based method, other methods have also been designed in which sensors are utilized to acquire gait data, including the number of footsteps, speed, step time, stance time, swing time, cadence, swing stance ratio, and stride interval-based features. These features are input into machine learning to classify healthy and abnormal gait patterns (*e.g.*, *Wahid et al. (2015)* and *Balaji, Brindha & Balakrishnan (2020)*). *Kwon et al. (2020)* employed medical data (radiographic imaging X-rays), including kinetic and kinematic gait data, to detect KOA. However, these methods (*Wahid et al., 2015*; *Kwon*

**Table 8 Comparative analysis with Existing Methods.**

| Authors | Technique | Vision based gait data | Both KOA and PD | Year | Dataset | Findings (Accuracy) |
|---|---|---|---|---|---|---|
| *Kour, Gupta & Arora (2022)* | KNN and FODPSO | ✓ | ✗ | 2022 | KOA-PD-NM Dataset | 90.00% |
| *Khessiba et al. (2023)* | Graph Convolutional Network | ✓ | ✗ | 2023 | KOA-PD-NM Dataset | 93.75% |
| *Wahid et al. (2015)* | RF and SVM | ✗ | ✗ | 2015 | Gait dataset (*Fernandez et al., 2013*) | 92.60% |
| *Balaji, Brindha & Balakrishnan (2020)* | DT, SVM, EC | ✗ | ✗ | 2020 | PhysioNet dataset (*Goldberger et al., 2000*) | 99.40% |
| *Kwon et al. (2020)* | Inception-ResNet-v2+SVM | ✗ | ✗ | 2020 | Own dataset (kinetic and kinematic gait data) | 75.50% |
| *Yang et al. (2020)* | Sensor based SVM | ✗ | ✗ | 2020 | Own dataset | 92.8% |
| *Buongiorno et al. (2019)* | Machine learning algorithms | ✗ | ✗ | 2019 | Own dataset | 81.00% |
| **Proposed** | **VGG16+GRU, MobileNet+GRU+ DenseNet+GRU** | ✓ | ✓ | **2025** | **KOA-PD-NM Dataset** | **94.81%** |

*et al., 2020*; *Balaji, Brindha & Balakrishnan, 2020*) were only investigated for PD or KOA. Compared with existing studies in Table 8, the proposed work is the extended version of identifying KOA, PD, and healthy gait patterns. Furthermore, the suggested technique is a vision-based approach operating on low-resolution videos, making this method cost-effective. The validity of the research is supported by different elements such as from existing literature, it is clearly concluded that human gait is disrupted by conditions such as Parkinson's disease (PD) and knee osteoarthritis (KOA). The study ensures the proper analysis of gait features by extracting silhouettes to acquire the spatiotemporal features from gait videos. The performance of the proposed model using different metrics indicates the worth of this study in diagnosing PD and KOA. Furthermore, the approach's focus on automated, non-invasive analysis makes it a viable alternative to established clinical approaches, highlighting its prospects for real-world use. The practical implications of this research include early detection and screening of patients with KOA and PD disease using a cost-effective, non-invasive, and automated approach. Early assessment, monitoring of patients, and rehabilitation planning might all be greatly improved by incorporating such a cost-effective deep learning-based model into clinical practice. The model can help clinicians diagnose KOA and Parkinson's disease by monitoring gait patterns before additional symptoms worsen. It might be particularly useful in telemedicine and remote health monitoring when patients do not have frequent access to professionals. Despite traditional gait analysis techniques, which necessitate motion-capturing systems or wearable sensors, the technique presented in this study is based simply on video analysis, making it more accessible and affordable. As a result, refining the model into a lightweight architecture that ensures consistent performance on low-resource devices is an important issue for future development. However, the trade-off between model accuracy and computing complexity remains an important consideration. In healthcare applications when patient health is at risk, accuracy takes priority over complexity, resulting in

dependable and accurate diagnostic support. Following that, data privacy and security are also critical since the system uses patient gait videos, which may be handled by implementing secure data storage, anonymization techniques, and adherence to healthcare rules. Federated learning-based algorithms are also a promising future avenue for increasing framework security. The proposed model may be connected with existing diagnostic tools such as electronic health records (EHRs) and decision-support systems, providing doctors with extra guidance and insight.

Rather than focusing just on the advantages, it is critical to discuss the limitations of this study for future researchers to expand on it. Hence, one possible limitation of this work is validation on a single small-scale dataset because there are no other publicly available vision-based gait datasets for diagnosing these abnormalities (KOA or PD). Second, the KOA–PD–NM dataset utilized in this research is also limited in the number of videos, camera viewpoints, and subjects, but it serves as a key foundation for future research in gait-based classification for KOA and PD detection. Particularly, the total number of PD videos is limited and leads to class imbalance and overfitting. Although, the dataset maintains a good diversity by considering different age groups, gender, and height, but still, a more diversified dataset is required to improve the performance. Hence, in the future, we aim to collect a large publicly available vision-based gait dataset to validate the results because the model's robustness and capacity to generalize across different patient groups would be improved with a larger and more diversified dataset. Similarly, more effective gait representations, including gait energy images or gait entropy images (*Bashir, Xiang & Gong, 2009*) with deep learning tactics, can also be explored. In addition, currently, the model is trained on specific labeled abnormalities *i.e.*, KOA and PD using the supervised learning concept, however, to generalize it well on other abnormalities, incorporating self-supervised learning or few-shot learning techniques could also be a future direction. Lastly, to enhance the model's robustness against adversarial attack-based perturbations (*i.e.*, noise), adversarial training can be incorporated as an additional defense mechanism.

## CONCLUSIONS

Gait is a synchronized, cyclic sequence of motions resulting in human locomotion and can be of great assistance in surveillance applications. However, various systematic disorders, such as injuries, PD, and KOA, can substantially influence human walking style. Hence, by analyzing gait patterns, it is feasible to determine whether the observed gait is abnormal and, in certain situations, to differentiate between distinct conditions that produce gait abnormalities. In existing studies, several methods have been suggested to address this problem; however, improving these methods by designing more advanced methods is a continual area of research, particularly in computer vision. Hence, we proposed an improved version of this analysis adopting a deep learning paradigm in which gait analysis is performed in two aspects, spatial and temporal, by running deep models in series. We proposed a transfer learning strategy with the CNN-RNN architecture by modifying the backbones (*i.e.*, VGG16, MobileNet, DenseNet, and GRUs). The proposed model was validated on the KOA–PD–NM dataset and achieved the highest accuracy of 94.81% with a

VGG16+GRUs model. In the future, we will extend this work by adding another module, which will also predict the severity of the abnormalities instead of classifying only the abnormalities.

### Funding
This research was supported by the Institute for Information & Communications Technology Planning & Evaluation (IITP)-ITRC (Information Technology Research Center) grant funded by the Korea Government (MSIT) (IITP-2025-RS-2024-00438056, 50%) and grant funded by the Korea Government (MSIT) (No. 2022R1F1A1063134, 50%). There was no additional external funding received for this study. The funders had no role in study design, data collection and analysis, decision to publish, or preparation of the manuscript.

### Grant Disclosures
The following grant information was disclosed by the authors:
Korea Government (MSIT): IITP-2025-RS-2024-00438056, 50% and 2022R1F1A1063134, 50%.

### Competing Interests
The authors declare that they have no competing interests.

### Author Contributions
- Zeeshan Ali conceived and designed the experiments, performed the computation work, prepared figures and/or tables, and approved the final draft.
- Jihoon Moon conceived and designed the experiments, analyzed the data, authored or reviewed drafts of the article, and approved the final draft.
- Saira Gillani conceived and designed the experiments, performed the computation work, prepared figures and/or tables, and approved the final draft.
- Sitara Afzal performed the experiments, performed the computation work, prepared figures and/or tables, and approved the final draft.
- Muazzam Maqsood performed the experiments, analyzed the data, authored or reviewed drafts of the article, and approved the final draft.
- Seungmin Rho performed the experiments, analyzed the data, authored or reviewed drafts of the article, and approved the final draft.

### Data Availability
The dataset is available at Mendeley: Kour, Navleen; Gupta, Sunanda; Arora, Sakshi (2020). "Gait dataset for knee osteoarthritis and Parkinson's disease analysis with severity levels", *Mendeley Data*, *V1*, DOI: 10.17632/44pfnysy89.1.
The source code is available in the Supplemental File.

## Supplemental Information

Supplemental information for this article can be found online at http://dx.doi.org/10.7717/peerj-cs.2857#supplemental-information.

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
