# Peer review of "Vision-based approach to knee osteoarthritis and Parkinson's disease detection utilizing human gait patterns"

_PeerJ Computer Science, doi:10.7717/peerj-cs.2857_

## Round 0.1 · original submission · Major Revisions

Dear authors,

You are advised to critically respond to all comments point by point when preparing an updated version of the manuscript and while preparing for the rebuttal letter. Please address all comments/suggestions provided by reviewers, considering that these should be added to the new version of the manuscript.

Kind regards,
PCoelho

Reviewer 1 ·

Basic reporting

The manuscript presents a study on the use of deep learning-based gait analysis for diagnosing musculoskeletal and neurological disorders, specifically knee osteoarthritis (KOA) and Parkinson’s disease (PD). The proposed approach, which leverages transfer learning and a CNN-RNN architecture, demonstrates significant advancements in both spatial and temporal gait pattern analysis. With a reported accuracy of 94.81% on the KOA–PD–normal dataset, the methodology showcases potential for clinical applications. While the study is practical, improvements in dataset description, comparative analysis, and result explainability would further enhance the work’s clarity and impact.

Experimental design

How does the dataset ensure sufficient diversity in terms of demographics (e.g., age, gender, ethnicity) and gait variability? Could potential biases in the dataset impact the generalizability of the proposed model to real-world scenarios?

Validity of the findings

Given the black-box nature of deep learning models, what steps have been taken to ensure the interpretability of the model’s decisions? Can you identify specific spatiotemporal features that the model relies on for classification?

Additional comments

1. While the proposed method achieves high accuracy, how does it compare to existing state-of-the-art techniques in terms of computational efficiency, robustness to noise, and ability to handle unseen abnormalities? Are there trade-offs between performance and complexity?

2. What are the practical implications of integrating the proposed model into a clinical setting? Have considerations been made for real-time processing, data privacy, or compatibility with existing diagnostic tools?

3. The reported results are impressive, but were they validated using cross-validation or an independent test set? Could overfitting be a concern, especially given the complexity of the model and the potential limitations of the dataset size?

·

Basic reporting

The authors presented a manuscript entitled “Vision-based approach to knee osteoarthritis and Parkinson’s disease detection utilizing human gait patterns.” The proposed model achieved the highest accuracy in classifying subjects with Parkinson’s disease, knee osteoarthritis, and healthy conditions based on gait patterns, reaching 94.81%.
The article was written clearly and organized well.

Experimental design

The details of the experimental design were provided in this manuscript.

Validity of the findings

They compared the proposed model with other models. Based on the data provided by the authors, the proposed model outperforms previously reported models. This study is impressive because it avoids the use of expensive sensors.

Additional comments

I have some comments on this manuscript that I kindly ask the authors to address and respond to appropriately:
1. Please use a consistent number of decimal places when presenting numerical values in your study.
2. The videos were filmed from a lateral view. Why were the gait videos not filmed from a front view? I suggest including images from the front view, as they may provide additional information and potentially enhance the model’s accuracy.
3. In line 125, the sentence should be revised to use the past tense for consistency.
4. In this study, the number of subjects with Parkinson’s disease is too small (only 16 cases). Could this affect the generalizability of the model? Please discuss and provide comments on this limitation.

Reviewer 3 ·

Basic reporting

The article is written in professional English with ease to understand.

Experimental design

Well design and Executed

Validity of the findings

Conclusion is properly explained.

Additional comments

More comparative result should be added from the existing research works.

---

## Round 0.2 · Minor Revisions

Dear authors,

Thanks a lot for your efforts to improve the manuscript.

Nevertheless, some concerns are still remaining that need to be addressed.

Like before, you are advised to critically respond to the remaining comments point by point when preparing a new version of the manuscript and while preparing for the rebuttal letter.

Kind regards,
PCoelho

·

Basic reporting

This study proposed a new approach in which spatiotemporal features are extracted via deep learning methods with a transfer learning paradigm. By using the proposed model, the gait patterns of KOA, PD, and healthy subjects could be distinguish with high accuracy. In addition, this model presented better accuracy compared with other existing models.

Experimental design

The research purpose is well-defined; however, the paper has a limitation on the dataset. Because the numbers of the video data with PD subjects in three severity levels are to few, especially for the severe level. For this severe group, there are only five videos. But, this dataset has also been adopted in other studies. Even though this is a shortcoming, I think it is still acceptable if we only focus on comparing the performance of models using the same data.

Validity of the findings

The manuscript quality requires improvement.
(1) In line 36 of the abstract, what is the full name of GRU? The full name of GRU must be provided here.
(2) In line 218, the authors stated they collected videos of healthy and PD or KOA patients. However, the dataset is still unknown. Until reading Line 327, the information about the dataset is revealed. I believe the numbers of healthy, and PD or KOA patients should be provided in the Materials & Methods section rather than in the Experiments and Discussion section.
(3) The format for referring to Figures is not consistent, like line 348. Figure or Fig.? Please check throughout the manuscript and revise the errors.
(4) Subfigures should be labeled. For example, I must distinguish the Figures 5(a), 5(b), 5(c), 5(d), 5(e), and 5(f) by myself. Please correct these errors in the manuscript.

Additional comments

I think the authors still need to make minor revisions to improve the manuscript's quality.

---

## Round 0.3 · accepted · Accept

Dear authors, we are pleased to verify that you meet the reviewer's valuable feedback to improve your research.

Thank you for considering PeerJ Computer Science and submitting your work.

Kind regards
PCoelho

·

Basic reporting

1. The manuscript is written in professional, clear, and unambiguous English. The grammar is generally strong and suitable for publication in an international academic journal.
2. The introduction provides sufficient background.
3. The study is well-supported by relevant and recent literature.
4. The introduction successfully contextualizes the medical relevance of gait abnormalities in KOA and PD and clearly outlines the motivation.

Experimental design

1. The authors conduct a rigorous technical investigation using well-established deep learning architectures (VGG16, MobileNet, DenseNet with GRUs).
2. The evaluation methods, assessment metrics, and model selection methods have been adequately described.
3. The dataset used (KOA–PD–Normal gait dataset) is publicly available, anonymized (with blurred faces), and ethically sound.

Validity of the findings

1. For the novelty of the approach, the study presents a novel integration of spatiotemporal gait analysis using vision-based data (silhouettes) with state-of-the-art CNN-GRU hybrid architectures.
2. Its ability to operate on low-resolution video makes it more accessible for resource-constrained environments, which is a big plus.
3. The use of a single dataset limits the study, and the PD class has relatively fewer samples.
4. The Conclusion states the performance of the proposed model and future directions.

Additional comments

None